# Machine Learning Approaches for the Prediction of Postoperative Major Complications in Patients Undergoing Surgery for Bowel Obstruction

**DOI:** 10.3390/jpm14101043

**Published:** 2024-10-08

**Authors:** Alessandro D. Mazzotta, Elisa Burti, Francesco Andrea Causio, Alex Orlandi, Silvia Martinelli, Mattia Longaroni, Tiziana Pinciroli, Tarek Debs, Gianluca Costa, Michelangelo Miccini, Paolo Aurello, Niccolò Petrucciani

**Affiliations:** 1Department of Surgery, Vannini General Hospital, Oncological and General Surgery, 00177 Rome, Italy; alex.mazzotta@gmail.com; 2The BioRobotics Institute, Sant’Anna School of Advanced Studies, 56127 Pisa, Italy; 3Department of Medical and Surgical Sciences and Translational Medicine, Division of General and Hepatobiliary Surgery, St. Andrea Hospital, Sapienza University of Rome, 00185 Roma, Italy; burti.1694089@studenti.uniroma1.it (E.B.); paolo.aurello@uniroma1.it (P.A.); niccolo.petrucciani@uniroma1.it (N.P.); 4Section of Hygiene, Department of Life Sciences and Public Health, Università Cattolica del Sacro Cuore, 00168 Rome, Italy; silvia.martinelli04@icatt.it; 5EIT Digital Master School, Polytech Nice Sophia, 06410 Biot, France; alex.orlandi@icloud.com; 6Department of Surgery, Santa Maria della Misericordia Hospital, University of Perugia, 06123 Perugia, Italy; longaronimattia@gmail.com; 7MIT Professional Education, Massachusetts Institute of Technology, Cambridge, MA 02139, USA; tiziana.pinciroli@gmail.com; 8Département de Chirurgie Digestive, Centre Hospitalier Universitaire de Nice, CHU Nice, 06000 Nice, France; dr.debs@hotmail.com; 9Department of Life Science, Health, and Health Professions, Link Campus University, 00165 Rome, Italy; g.costa@unilink.it; 10Department of Surgery, Sapienza University of Rome, 00185 Roma, Italy; michelangelo.miccini@uniroma1.it

**Keywords:** small bowel obstruction, complication, outcome, survival

## Abstract

Background: Performing emergency surgery for bowel obstruction continues to place a significant strain on the healthcare system. Conventional assessment methods for outcomes in bowel obstruction cases often concentrate on isolated factors, and the evaluation of results for individuals with bowel obstruction remains poorly studied. This study aimed to examine the risk factors associated with major postoperative complications. Methods: We retrospectively analyzed 99 patients undergoing surgery from 2015 to 2022. We divided the patients into two groups: (1) benign-related obstruction (n = 68) and (2) cancer-related obstruction (n = 31). We used logistic regression, KNN, and XGBOOST. We calculated the receiver operating characteristic curve and accuracy of the model. Results: Colon obstructions were more frequent in the cancer group (*p* = 0.005). Operative time, intestinal resection, and stoma were significantly more frequent in the cancer group. Major complications were at 41% for the cancer group vs. 20% in the benign group (*p* = 0.03). Uni- and multivariate analysis showed that the significant risk factors for major complications were cancer-related obstruction and CRP. The best model was KNN, with an accuracy of 0.82. Conclusions: Colonic obstruction is associated with tumor-related blockage. Malignant cancer and an increase in C-reactive protein (CRP) are significant risk factors for patients who have undergone emergency surgery due to major complications. KNN could improve the process of counseling and the perioperative management of patients with intestinal obstruction in emergency settings.

## 1. Introduction

Bowel obstruction (BO) represents one of the most common acute abdominal conditions requiring emergency surgery, accounting for approximately 15% of hospital admissions for acute abdominal pain in the United States and 20% of cases needing acute surgical care [1]. Small bowel obstructions are caused in 90% of cases by adhesions, hernias, and neoplasms, whereas large bowel obstructions are provoked by cancer in about 60% of cases; volvulus and diverticular disease are responsible for the other 30% [2,3].

This represents a significant burden on healthcare systems due to the disease severity, substantial morbidity and mortality rates, and associated economic costs [4,5]. Most patients presenting with bowel obstruction require urgent evaluation, primarily relying on contrast-enhanced Computed Tomography (CT) imaging [6]. Recent research has explored the utility of serum procalcitonin as a potential biomarker for predicting the failure of conservative management in small bowel obstruction cases as well as an indicator of intestinal ischemia, a particularly challenging complication to diagnose [7].

Numerous clinical scoring systems aim at predicting bowel ischemia with the aim of identifying patients with more severe diseases who require emergency surgical intervention or those at high risk of non-operative management failure; however, their validity varies across diverse clinical contexts and geographical regions, thus restricting their generalizability and making bowel obstruction management a significant challenge in clinical practice [8]. Novel approaches typically integrate a combination of clinical and laboratory data or incorporate radiological findings. However, the efficacy and reliability of these integrated scoring systems also warrant further investigation and validation [9,10] The decision-making process regarding the necessity and timing of surgical intervention for BO, as well as the prediction of stoma creation or bowel resection, largely relies on clinicians’ subjective assessments, often characterized as “gut-feeling” judgments. Reliable and objective predictive tools would enhance patient counseling and optimize postoperative care strategies, potentially reducing the variability in patient outcomes [11]. Artificial intelligence (AI) is increasingly used in other fields, such as radiology [12] or clinical trial design [13]. In emergency settings, several applications are being tested where AI-based solutions can assist, e.g., in triaging patients or in predicting disease progression [14], but little research has been conducted on its implementation in bowel obstruction. The present study aims to analyze the role of machine learning algorithms in bowel obstruction and their ability to predict postoperative complications.

## 2. Materials and Methods

### 2.1. Study Design

This investigation was a component of an international collaborative research program involving undergraduate students from Italian and French universities. The study design was observational and retrospective, utilizing data from a prospectively maintained database focused on emergency and trauma surgery at the Sant’Andrea Sapienza University teaching hospital in Rome, Italy. The establishment and maintenance of this database were conducted in accordance with previously approved research protocols and studies by the relevant ethical committee [15,16].

This adherence to established ethical guidelines ensured the integrity and validity of the data used in the current study. This research framework allowed for the analysis of real-world clinical data while fostering international academic collaboration at the undergraduate level. For this study, formal institutional review board approval was not required due to the study design; however, we obtained signed consent for the storage and analysis of data for scientific purposes from all patients at admission.

### 2.2. Population and Data

Medical records of patients who underwent surgery for bowel obstruction between January 2019 and December 2022 were retrieved. The inclusion criteria were as follows: (1) patients over 18 years of age; (2) admission through the emergency department; (3) intraoperative confirmed diagnosis of bowel obstruction; and (4) procedures performed or directly supervised by senior surgeons. Patients were excluded if they met the following criteria: (1) they were already hospitalized; (2) they were already scheduled for elective surgery; (3) they were patients with intra-abdominal sepsis such as appendicitis, acute cholecystitis, or acute diverticulitis; (4) they were patients with peritoneal free air; or (5) they were patients participating in other randomized or interventional clinical trials.

To reduce bias, we excluded the procedures that were not performed or tutored by two senior surgeons (PA, NP). After the identification of eligible patients, we collected data including demographic characteristics and clinical variables, level and type of obstruction, procedure details, and outcomes. Demographics variables and clinical data included the following: age, gender, weight, height, body mass index (BMI), heart rate, systolic blood pressure, medical and surgical history (comorbidities), and common preoperative biochemical blood examination (including C-reactive protein [CRP] and arterial blood gas analysis). Causes of bowel obstruction were divided as neoplastic, adhesions, sigmoid volvulus, and abdominal wall hernia either primary or incisional. Comorbidity was recorded if the condition was present at admission. Preoperative risk was assessed with the American Society of Anaesthesiologists (ASA) score. The Age-adjusted Charlson Comorbidity Index (age-CACI) was also calculated [17].

Intensive Care Unit (ICU) admission, length of ICU stay, and length of hospital stay (LOS) were recorded. Morbidity and mortality were considered as the 30-day standard period definition. Any adverse outcomes were also considered regardless of the time elapsed if they occurred during the hospitalization following the index emergency procedure. Postoperative complications were classified according to the Clavien–Dindo system [18]. Grade IIIa or higher complications were considered as major complications. No enhanced recovery after surgery (ERAS) protocol was adopted, but in the postoperative period, the patients were managed following the same standardized pathways of care both for uneventful and complicated courses. Data about the postoperative outcome was available for all patients.

All pre-operative characteristics, such as clinical and demographic, level of occlusion, and radiological bowel findings, along with post-operative data, including minor and major complications, were blind-reviewed and coded by two expert surgeons (MM, TD). Any disagreements were discussed and resolved through a consensus meeting with a third senior emergency surgeon (GC). Finally, data were entered into a new specific worksheet crafted with LibreOffice (Vers. 7.6.7 for Windows). Table 1 shows a list of all the investigated variables and their descriptions.

### 2.3. Machine Learning Methodology

The machine learning study was carried out by A.O and T.P., leveraging models that included algorithms based on supervised learning. A sample of 70% of the cohort generated randomly using a seed was applied for the training set; the remaining 30% was used for testing. Compared to a 50–50 division, a 70% training split provides the models with more information, compared to using 50%, where the models could be under-trained. In addition to this, allocating more data for training helps the models to reduce variance and capture the underlying patterns in a more efficient way. Based on the predictive factors selected, 3 models were constructed including support K-nearest neighbor (KNN), XgBoost, and logistic regression. During training, the algorithm iteratively learns the optimal path from root nodes to leaves by minimizing prediction errors. This path signifies the classification rules guiding the decision model’s predictions for new patients. KNN builds a robust predictive model by aggregating information from multiple neighbors, thus mitigating the risk of overfitting associated with highly complex models.

Logistic regression is a supervised learning algorithm used for binary classification tasks. It models the probability that a given input belongs to a particular class using a sigmoid function to map predicted values to probabilities between 0 and 1. XGBoost is a scalable machine learning algorithm that uses an ensemble of decision trees, built sequentially, to improve prediction accuracy by minimizing errors from previous models through gradient boosting techniques. KNN is a simple algorithm that classifies data points based on the majority class of their nearest neighbors in the feature space. Here, “k” represents the number of neighbors to consider. It has the ability to handle missing values by defining default directions for each node in case of missing data. To train and validate the performance of the KNN algorithm, we employed repeated stratified ten-fold cross-validation, repeating the process five times. This method ensured each patient appeared at least once in the testing set, cycling the dataset into ten equally sized folds for training and validation. In our study, we optimized the KNN hyperparameters through a grid search and five-fold cross-validation. The grid search space included parameters such as ‘n_neighbors’ and the distance metric, allowing us to identify systematically the optimal configuration for the KNN algorithm. Feature selection using the coefficients of logistic regression was applied. This involved the identification of the most important features based on their impact on the model’s predictions. In logistic regression, the model assigns a coefficient to each feature, representing how strongly that feature influences the predicted outcome. Features with larger absolute coefficients are considered more important because they have a greater effect on the target variable. By analyzing these coefficients, it is possible to select the most relevant features for the model. The models were evaluated and compared by sensitivity, specificity, and the area under the curve (AUC) of the receiver operating characteristic (ROC) curve. The analyses were performed using Python 3.12.

### 2.4. Statistical Analysis

Statistical analysis was carried out using IBM Corp, released 2013, IBM SPSS Statistics for Windows, Version 22.0. Armonk, NY, USA: IBM Corp. Dichotomous data and counts were presented in frequencies, whereas continuous data were presented as mean values ± standard deviations (SDs). Differences between means were compared using Student’s independent sample *t*-test, the Mann–Whitney U test, or the Kruskall–Wallis test when indicated. Fisher’s exact test or χ^2^ test, with or without the Yates correction, were used to compare differences in frequencies. Variables significantly associated with major postoperative morbidity were identified using Cox’s univariate and multivariate analyses. The level of significance to allow for the inclusion of a variable in the logistic regression multivariate model was 0.05 for major complications. Any statistical significance was set at *p* < 0.05.

## 3. Results

Ninety-nine patients undergoing emergency surgery for bowel obstruction were included. Figure 1 illustrates the participant selection and allocation process for the AI algorithm training and validation datasets. A flow chart of the patients is provided in Appendix A.

Table 1 reports the general characteristics of the population; data are reported for the overall population and according to the benign or malignant etiology of the obstruction. The majority of patients had small bowel obstruction (63.6%), whereas colonic obstruction was present in 32% of patients. The majority of patients had previous abdominal surgery (62.3%). The median BMI was 23.6 [20.7–27.6]. C-reactive protein was above normal values (>1.0 mg/100 mL) in 60% of cases. The average waiting time between admission and abdominal CT scan was 10 h. At CT scan, fluid effusion was detected in 59.2% of patients and bowel ischemia in 12.2% of cases.

A laparoscopic approach was attempted in 34 patients, 18 of which were converted to open surgery. The mean operative time was 105 min. Histological exams were performed in 43 cases, and in 24 cases, the diagnosis of malignant disease was confirmed.

Table 2 reports intraoperative and postoperative outcomes. The major complication rate was 27.3% for all patients, with a significant difference between benign obstruction and malignant obstruction (20% vs. 41%, respectively). Postoperative death occurred in seven patients. The mean hospital stay was 10 days.

The univariate and multivariate analyses showed that CRP (OR 1.18, CI 1.05–1.3, *p* = 0.003) and cancer-related obstruction (OR 4.2, CI 1.2–14.0, *p* = 0.02) were independent risk factors for major complications (Table 3).

We compared the capacity of XGBOOST, KNN, and logistic regression to predict the outcome with good performance, as shown in Figure 1. KNN was considered the best algorithm, with an area under the curve (AUC) of 0.77 and an overall accuracy of 0.88. The most significant features were distal bowel collapse with an importance value of 0.92, followed by ASA > 2 at 0.75 and creatinine at 0.6. An operative time > 105 min and mesentery edema are also notable, with importance values of 0.52 and 0.45, respectively. Figure 2 shows a gradual decrease in importance for subsequent features.

## 4. Discussion

Bowel obstruction represents a significant challenge for health systems worldwide. It causes 15% of emergency department access for abdominal pain [1], and it has different causes and possibly presents as heterogeneous clinical scenarios. Morbidity and mortality rates are elevated, particularly in patients who need emergency surgery [19,20,21]. The complexity and heterogeneity of bowel obstruction makes the prediction of the outcomes very difficult for clinicians. Even patients undergoing surgery for bowel obstruction due to adherences (one of the less “severe” scenarios) may experience intraoperative complications such as bowel perforations and postoperative major complications. A rapid analysis of patient characteristics and preoperative exams is the responsibility of the on-call surgeons to assess the need of surgery and type of approach [22] to predict potential outcomes in order to correctly inform the patient and eventually their family and to program the most appropriate setting for postoperative surveillance.

Artificial intelligence (AI) represents a promising tool to aid physicians in completing complex tasks. In the setting of small bowel obstruction, previous research has shown the ability of neural networks to identify small bowel obstruction on plain radiographs with a high degree of accuracy [23,24].

AI could represent a valid tool to integrate a large amount of clinical and radiological information and predict postoperative complications and to facilitate the decision-making process of surgeons giving an indication for surgery and to provide correct patient counseling. The World Society of Emergency Surgery has included bowel obstructions among the clinical scenarios where neural networks may support clinical decisions [24].

The present study demonstrated that machine learning was able to predict the occurrence of postoperative complications with an overall accuracy of 0.88, performing better than the available clinical risk score. As shown in Figure 2, the AI-based model was able to classify the different variables according to their importance in predicting the outcomes. Among the variables with higher predictive value for postoperative major morbidity, preoperative C-reactive protein (CRP) and malignant obstruction were identified. C-reactive protein nowadays represents a useful tool for the early detection of postoperative complications after several abdominal procedures [23,25].

In the setting of emergency surgery, preoperative CRP was associated with a higher incidence of difficult cholecystectomies and perforated appendicitis [26,27], but its role in predicting the outcomes after surgery for bowel obstruction has not yet been reported.

The other significant factors were the malignant nature of the obstruction, confirming the results of previous studies showing the worst prognosis of malignant obstructions in terms of postoperative complications and need of stoma creation [28]. Out of 20 stomas created, 14 were carried out in patients with cancer, and only 3 of them were reconstructed during follow-up. In addition, the use of laparoscopy was concordant with previous reports [29,30]. In our series, the laparoscopic approach was effective in 16 cases only, whereas in 18 cases, open conversion was needed. A laparoscopic approach reduces the overall morbidity and length of stay and facilitates postoperative recovery [31,32,33], but segments lacking surgical pace due to an overdistended colon or a small bowel and the difficult manipulation of dilated segments often limit its feasibility. The analysis of large quantities of data with AI could, in the future, allow for identifying patients at greater risk of major complications and modulating the clinical approach by dedicating greater resources and establishing dedicated paths for patients at greater risk of major complications. A detailed collection of international multicenter data that flow into a single dataset is desirable.

### Limitations

This study has several limitations to consider when interpreting the results. Its retrospective design introduces potential biases in data collection and analysis. Being a single-center experience from Sant’Andrea Sapienza University teaching hospital in Rome, Italy, the findings may have limited generalizability to other healthcare settings or regions. The relatively small sample size of 99 patients, particularly when divided into subgroups, may limit its statistical power and the ability to detect smaller effect sizes. Examining small bowel obstruction and large bowel obstruction together may limit the specificity of our findings, and future studies should consider focusing on individual pathologies to provide more targeted insights.

This study’s time span from 2019 to 2022 could introduce variability due to evolving clinical practices. The exclusion of certain patient groups limits the applicability of findings to those populations. The machine learning models, while promising, lack external validation, which is crucial for assessing their true predictive performance and generalizability. There was limited direct comparison with specific, widely used clinical risk scores for bowel obstruction. Given the small dataset and complex machine learning models, there is a risk of overfitting, potentially leading to optimistic performance estimates. Lastly, the focus on 30-day morbidity and mortality may not capture longer-term outcomes relevant to bowel obstruction patients, particularly those with cancer-related obstructions. These limitations highlight the need for larger, multicenter prospective studies with external validation to confirm the predictive value of the machine learning approaches in patients undergoing surgery for bowel obstruction.

## 5. Conclusions

This study demonstrates the potential of machine learning approaches, particularly the K-nearest neighbors (KNN) algorithm, in predicting major postoperative complications for patients undergoing emergency surgery for bowel obstruction. Our model achieved an overall accuracy of 0.88 and an area under the curve (AUC) of 0.77, outperforming traditional clinical risk scores. The analysis identified preoperative C-reactive protein (CRP) levels and cancer-related obstruction as significant independent risk factors for major complications. The ability to rapidly integrate multiple clinical and radiological variables could enhance the precision of predicting postoperative complications, facilitating more informed surgical decisions and appropriate postoperative care planning.

## Figures and Tables

**Figure 1 jpm-14-01043-f001:**
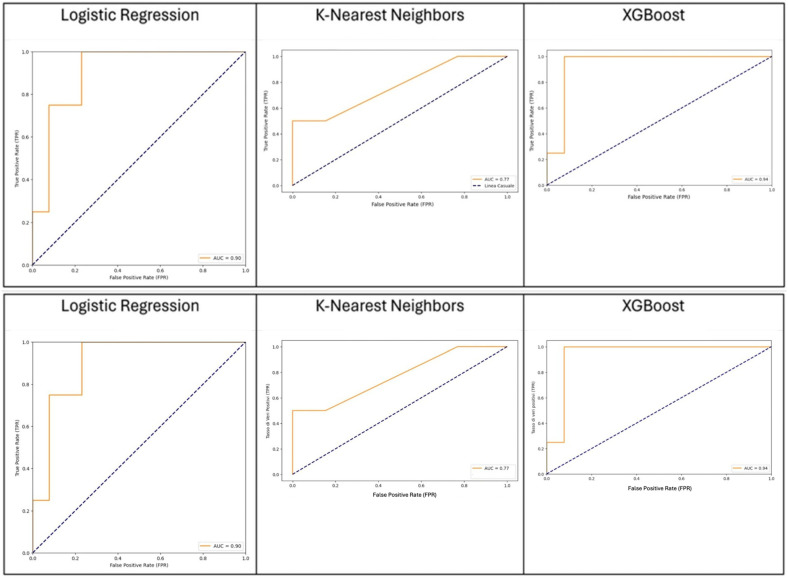
Comparative performance of XGBoost, KNN, and logistic regression models in predicting the outcome.

**Figure 2 jpm-14-01043-f002:**
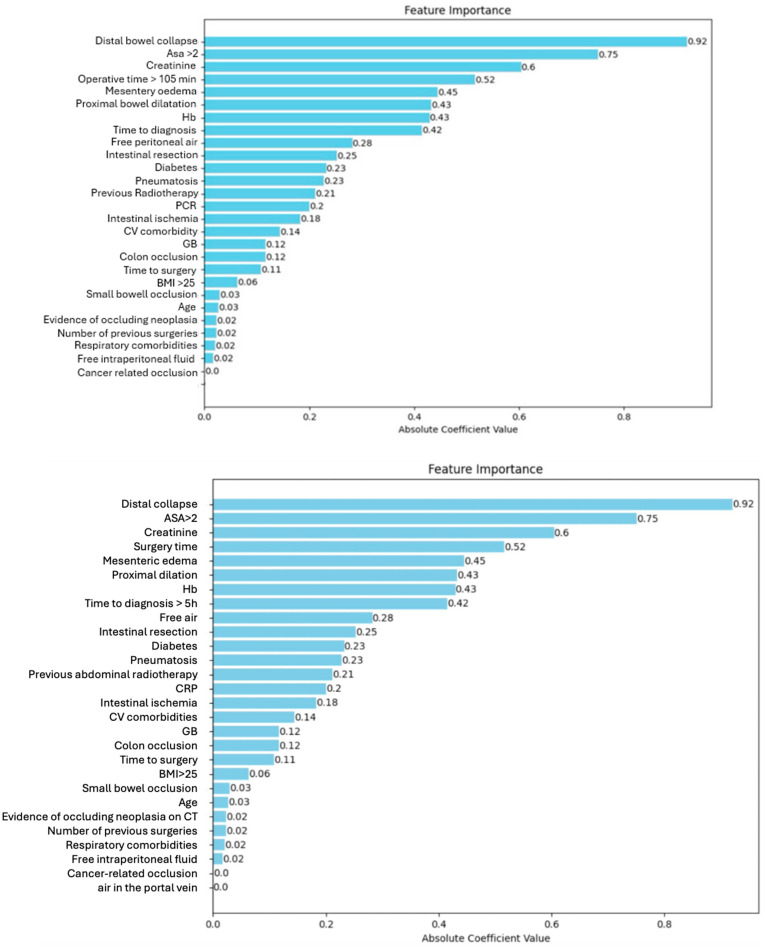
Feature importance.

**Table 1 jpm-14-01043-t001:** Patients’ demographics.

	Total n = 99	Benign n = 65	Cancer n = 34	*p* Value
Age	69 [58–79]	69 [56–79]	71 [57–78]	0.60
Sex (M)	52 (52.3)	31 (47.7)	21 (61.8)	0.20
ASA > 2	66 (66)	39 (97.5)	96.4 (27)	0.80
BMI	23.6 [20.7–27.6]	23.4 [20.6–28.1]	24.0 [20.8–26.7]	0.90
Lung comorbidity	6 (6.1)	6 (9)	0 (0)	0.07
CV comorbidity	45 (45.5)	29 (44.6)	16 (47.1)	0.80
Colon occlusion	32 (32)	8 (12.3)	24 (70.6)	0.005
Small Bowell occlusion	63 (63.6)	56 (86.2)	7 (20.6)	0.005
Previous surgery	62 (62.6)	44 (67.7)	18 (52.9)	0.15
CT sign of ischemia	12 (12.2)	12 (18.5)	0 (0)	0.005
Free peritoneal fluid	58 (59.2)	37 (56.9)	21 (63.6)	0.52
Hemoglobin	14.1 [12.4–15.4]	14.4 [12.8–15.4]	13.05 [11.2–14.7]	0.01
WBC	10.3 [7.7–14]	10.5 [7.7–13.4]	10.2 [7.3–15.1]	0.72
CRP	1.0 [0.3–5.3]	1.0 [0.3–4.6]	1.49 [0.3–5.5]	0.63

n: number; [ ]: Interquartile range; ( ): percentage; CV = cardiovascular; CT = Computed Tomography; WBC = White Blood Cells; CRP = C-reactive protein.

**Table 2 jpm-14-01043-t002:** Intraoperative results and post-operative outcomes.

	Total n = 99	Benign n = 65	Cancer n = 34	*p* Value
Operative time	105 [80–175]	95 [75–141]	150 [95–205]	0.001
Intestinal resection	43 (43.4)	20 (30.8)	23 (67.6)	0.005
Stoma	20 (20.6)	6 (9.4)	14 (42.4)	0.005
Major complications	27 (27.3)	13 (20)	14 (41)	0.03
Kidney failure	4 (4.3)	3 (4.9)	1 (3.2)	0.70
Surgical site infection	5 (5.4)	2 (3.3)	3 (9.7)	0.20
Hospital stay length	10 [7.5–18]	9 [6–14]	15.5 [8.7–21]	0.001

n: number; [ ]: interquartile range; ( ): percentage.

**Table 3 jpm-14-01043-t003:** Univariate and multivariate analyses for major complications.

Univariate Analysis				Multivariate Analysis		
	OR	CI	*p* Value	OR	CI	*p* Value
ASA > 2	0.60	0.03–9.5	0.70			
Age	1.04	1.0–1.18	0.01	1.03	0.98–1.08	0.19
BMI > 25	0.08	0.34–2.25	0.80			
Previous surgery	0.70	0.27–1.60	0.37			
Small bowel occlusion	0.44	0.18–1.09	0.08			
Colon occlusion	1.5	0.61–3.80	0.35			
Cancer-related occlusion	2.8	1.12–6.98	0.03	4.2	1.2–14.0	0.02
CV comorbidity	1.42	0.58–3.4	0.43			
Lung comorbidity	2.8	0.54–15.2	0.21			
WBC	1.13	1.03–1.22	0.01	1.11	0.97–1.2	0.11
CRP	1.18	1.0–1.3	0.002	1.18	1.05–1.3	0.003
CT free peritoneal fluid	1.01	0.40–2.4	0.90			
CT ischemia	0.86	0.21–3.4	0.80			
Operative time (>105 min)	2.23	0.88–5.6	0.08			
Intestinal resection	1.3	0.53–3.16	0.60			

BMI = body mass index; WBC = White Blood Cells; CV = cardiovascular; CRP *=* C-reactive protein; CT = Computed Tomography.

## Data Availability

Data are not available due to the privacy policy.

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
