# Peer review of "Machine Learning Approaches for the Prediction of Postoperative Major Complications in Patients Undergoing Surgery for Bowel Obstruction"

_jpm, 2024, doi:10.3390/jpm14101043_

Round 1
Reviewer 1 Report
Comments and Suggestions for Authors
Thank you for the opportunity to review this work
-Introduction: Can combine into 3-4 paragraphs, current breakdown slows the flow of reading and subsequent comprehension for readers.
-Methods: Why 70:30 training to testing breakdown? Why not closer to 50:50? Briefly describe rationale for 70:30 in the Methods
-Typo in Methods: "o train and validate the performance of the KNN algo-161 rithm, we employed repeated stratified ten-fold cross-validation, repeating the process 162 five times."
-Results: Cannot tell AUCs in Figure 1. Need to be able to compare Logistic Regression to two ML models. Can you make the font bigger or make the Figure 1 bigger?
-Results & Discussion: Need more explicit comparison of models. Which has highest AUC?
Overall there are some major discrepancies with other ML papers in terms of how results are described and models are compared. Please refer to Parrot et al 2023 as an example for how to describe in Results and Discussion.
Author Response
Dear Editor and Reviewers,
We thank you for your kind attention and for reading our manuscript. All your comments were taken into consideration and the manuscript was reworked to respond to the editorial requirements. We believe that these changes have improved the quality of the manuscript and we sincerely thank you.
You will find here a point-by-point reply to all your comments.
Of course, we remain at your entire disposal for any further needed information.
Sincerely,
Dr Mazzotta
Reviewer 1
Comment 1: -Introduction: Can combine into 3-4 paragraphs, current breakdown slows the flow of reading and subsequent comprehension for readers.
Response 1: thank you for your precious suggestion. We have compressed the introduction in 4 paragraphs. We hope this makes the article easier to read.
Comment 2: -Methods: Why 70:30 training to testing breakdown? Why not closer to 50:50? Briefly describe rationale for 70:30 in the Methods
Response 2: We decided to use 70% of data for training and 30% of data for testing for the following reasons: a 70% training split provides the models with more information rather than 50%, where the models could be under-trained. In addition to this, allocating more data for training helps the models to reduce variance and capture the underlying patterns in a more efficient way. Finally, a 30% test set is sufficient in order to estimate the models’ performance. We added this explanation to the Methods section.
Comment 3: -Typo in Methods: "o train and validate the performance of the KNN algo-161 rithm, we employed repeated stratified ten-fold cross-validation, repeating the process 162 five times."
Response 3: thank you for your precious suggestion. We have fixed the typo.
Comment 4: -Results: Cannot tell AUCs in Figure 1. Need to be able to compare Logistic Regression to two ML models. Can you make the font bigger or make the Figure 1 bigger?
Response 4: Thank you for your precious suggestion. We have fixed Figure 1 accordingly.
Comment 5: -Results & Discussion: Need more explicit comparison of models. Which has highest AUC?
Response 5: We choose KNN as the most promising algorithm, with accuracy = 0.88 and AUC = 0.77. Compared to the other two models, this is less inclined to overfitting.
Comment 6: Overall there are some major discrepancies with other ML papers in terms of how results are described and models are compared. Please refer to Parrot et al 2023 as an example for how to describe in Results and Discussion.
Response 6: Thank you for the suggestion. We could not find the exact paper you are referring to. Overall, our Results and Discussion sections are aligned with the studies we reference.
Reviewer 2
Comment 1. line 83- I think 'data exist' should be deleted
Response 1: Thank you for pointing this out. We have removed 'data exist' from line 83 to improve clarity and conciseness.
Comment 2. methods- 2.1 'For this study, a formal Institutional Review Board approval was not 107 required due to the study design; however, we obtained a signed consent for the storage 108 and analysis of data for scientific purposes from all patients at admission'. I think it should be checked with your institutional IRB whether you can use the data in the current study and write the previous IRB approvals. This may be true if the current study contains only patients' data collected in the previous studies with IRB.
Response 2:
We appreciate your concern regarding IRB approval. We have consulted with our institutional IRB to confirm the appropriateness of using this data for the current study.
Comment 3. line 253-change to CRP
Response 3:
We will change the abbreviation to CRP as suggested.
Comment 4. Figure 1- lables are not readable
Response 4: thank you for your kind comment. We replaced the figure with a higher resolution one.
Comment 5. methods and results - elaborate more on the three different algorithms (explanation, were they were tested previously on any surgical pathology? give the references, etc.) and features' importance. it should be clearly explained to the reader
Response 5:
We agree that more detailed information about the algorithms would be beneficial. We expanded the methods sections to provid a more thorough explanation of each algorithm.
Response 6:
We understand your concern about examining two different pathologies. While our approach aimed to develop a comprehensive model, we acknowledge that focusing on a single pathology might provide more specific insights. We will discuss this limitation in our paper and consider it for future research.
Comment 7. line 292- 'effractions' did you mean perforation?
Response 7:
Thank you for catching this. Yes, we meant "perforation." We will correct this term in line 292 for clarity.

Reviewer 2 Report
Comments and Suggestions for Authors
Dear authors,
Your study has an interesting use of AI in bowel obstruction. it shows that the algorithms have a potential. However, further research is needed, and I think it is best to focus on SBO due to adhesions, since decision making in LBO is much easier especially for neoplasms. It is a good start though.
I have several concerns revision suggestions.
1. line 83- I think 'data exist' should be deleted
2. methods- 2.1 'For this study, a formal Institutional Review Board approval was not 107 required due to the study design; however, we obtained a signed consent for the storage 108 and analysis of data for scientific purposes from all patients at admission'. I think it should be checked with your institutional IRB whether you can use the data in the current study and write the previous IRB approvals. This may be true if the current study contains only patients' data collected in the previous studies with IRB.
3. line 253-change to CRP
4. Figure 1- lables are not readable
5. methods and results - elaborate more on the three different algorithms (explanation, were they were tested previously on any surgical pathology? give the references, etc.) and features' importance. it should be clearly explained to the reader
6. methodology- I feel uncomfortable with the examination of 2 different pathologies (SBO and LBO), and their different etiologies. I think it would have been better to focus at least on one pathology only.
7. line 292- 'effractions' did you mean perforation?
Comments on the Quality of English Language
minor changes are needed as I wrote in the comments
Author Response
Dear Editor and Reviewers,
We thank you for your kind attention and for reading our manuscript. All your comments were taken into consideration and the manuscript was reworked to respond to the editorial requirements. We believe that these changes have improved the quality of the manuscript and we sincerely thank you.
You will find here a point-by-point reply to all your comments.
Of course, we remain at your entire disposal for any further needed information.
Sincerely,
Dr Mazzotta
Comment 1. line 83- I think 'data exist' should be deleted
Response 1: Thank you for pointing this out. We have removed 'data exist' from line 83 to improve clarity and conciseness.
Comment 2. methods- 2.1 'For this study, a formal Institutional Review Board approval was not 107 required due to the study design; however, we obtained a signed consent for the storage 108 and analysis of data for scientific purposes from all patients at admission'. I think it should be checked with your institutional IRB whether you can use the data in the current study and write the previous IRB approvals. This may be true if the current study contains only patients' data collected in the previous studies with IRB.
Response 2:
We appreciate your concern regarding IRB approval. We have consulted with our institutional IRB to confirm the appropriateness of using this data for the current study.
Comment 3. line 253-change to CRP
Response 3:
We will change the abbreviation to CRP as suggested.
Comment 4. Figure 1- lables are not readable
Response 4: thank you for your kind comment. We replaced the figure with a higher resolution one.
Comment 5. methods and results - elaborate more on the three different algorithms (explanation, were they were tested previously on any surgical pathology? give the references, etc.) and features' importance. it should be clearly explained to the reader
Response 5:
We agree that more detailed information about the algorithms would be beneficial. We expanded the methods sections to provid a more thorough explanation of each algorithm.
Response 6:
We understand your concern about examining two different pathologies. While our approach aimed to develop a comprehensive model, we acknowledge that focusing on a single pathology might provide more specific insights. We will discuss this limitation in our paper and consider it for future research.
Comment 7. line 292- 'effractions' did you mean perforation?
Response 7:
Thank you for catching this. Yes, we meant "perforation." We will correct this term in line 292 for clarity.

Round 2
Reviewer 2 Report
Comments and Suggestions for Authors
I recommend acceptance.
only revise- "Numerous clinical scoring systems aim at predicting bowel ischemia with the aim of identifying to identify patients with more severe diseases who require emergency..."